# Molecular Structure of Gaseous Oxopivalate Co(II): Electronic States of Various Multiplicities

**DOI:** 10.3390/ijms241713224

**Published:** 2023-08-25

**Authors:** Nina I. Giricheva, Valery V. Sliznev, Andrey S. Alikhanyan, Ekaterina A. Morozova, Georgiy V. Girichev

**Affiliations:** 1Nanomaterial Research Institute, Ivanovo State University, Ermak Str. 39, 153025 Ivanovo, Russia; n.i.giricheva@mail.ru; 2Department of Physics, Ivanovo State University of Chemistry and Technology, Sheremetevsky Ave. 7, 153000 Ivanovo, Russia; sliznev@mail.ru; 3Kurnakov Institute of General and Inorganic Chemistry RAS, Leninsky Prospect 31, 119991 Moscow, Russia; alikhan@igic.ras.ru (A.S.A.); cathrine_15@mail.ru (E.A.M.)

**Keywords:** polynuclear complex, molecular structure, electronic states, multiplicity, gas electron diffraction

## Abstract

Synchronous electron diffraction/mass spectrometry was used to study the composition and structure of molecular forms existing in a saturated vapor of cobalt(II) oxopivalate at T = 410 K. It was found that monomeric complexes Co_4_O(piv)_6_ dominate in the vapor. The complex geometry possesses the C_3_ symmetry with bond lengths Co–O_c_ = 1.975(5) Å and Co–O = 1.963(5) Å, as well as bond angles O_c_–Co–O = 111.8(3)°, Co–O_c_–Co = 110.4(6)°, O–Co–O = 107.1(3)° in the central O_c_Co_4_ fragment and four O_c_CoO_3_ fragments. The presence of an open 3d shell for each Co atom leads to the possibility of the existence of electronic states of the Co_4_O(piv)_6_ complex with Multiplicities 1, 3, 5, 7, 9, 11, and 13. For them, the CASSCF and XMCQDPT2 calculations predict similar energies, identical shapes of active orbitals, and geometric parameters, the difference between which is comparable with the error of determination by the electron diffraction experiment. QTAIM and NBO analysis show that the Co–O_c_ and Co–O bonds can be attributed to ionic (or coordination) bonds with a significant contribution of the covalent component. The high volatility and simple vapor composition make it possible to recommend cobalt (II) oxopivalate as precursors in the preparation of oxide films or coatings in the CVD technologies. The features of the electronic and geometric structure of the Co_4_O(piv)_6_ complex allows for the conclude that only a very small change in energy is required for the transition from antiferromagnetically to ferromagnetically coupled Co atoms.

## 1. Introduction

The method of chemical vapor deposition (CVD) makes it possible to obtain uniform films with good reproducibility and adhesion at reasonably high deposition rates and is widely used to obtain functional materials based on metal, oxide, carbide, and other coatings [1]. In addition to organometallic compounds, great opportunities open the use of metal complexes compounds with organic ligands–β-diketonates, carboxylates, etc. [2,3,4,5,6,7]. Metal carboxylates are an extensive class of coordination compounds. The wide possibilities of structural functions of acid residues provide a variety of structures of carboxylate complexes [8,9,10,11]. In recent decades, the salts of trimethylacetic (pivalic) acid, which are highly volatile, and congruently sublimating compounds have been actively studied.

The use of cobalt pivalate complexes as suitable precursors in the CVD technique is promising to produce multifunctional property coatings and materials based on cobalt oxides CoO and Co_3_O_4_ [12,13,14]. For example, cobalt monoxide CoO can be used as a colored admixture for glass and ceramics; it exhibits magnetic, catalytic, and sensing properties [15,16,17]. Most stable cobalt oxide Co_3_O_4_ possessing semiconductor properties can be used as electrode material in supercapacitors [15,16,17] as material for the anode of Li–ion batteries [18].

One of the representatives of such complexes is the cobalt(II) oxotrimethylacetate complex. The synthesis of this compound was first described in 1966 [19]. The author reports that the crystalline unit cell is cubic, space group Pa-3, and contains eight tetranuclear complexes [Co_4_(µ_4_-O)(µ_2_–OOCCMe_3_)_6_] [hereinafter Co_4_O(piv)_6_, where piv=OOC–CMe_3_] of C_3_ symmetry, and the compound has a low magnetic susceptibility, which is unusual for tetrahedrally coordinated Co atoms.

In [20,21], the synthesis of an oxotrimethylacetate complex of cobalt(II) was performed. There are contradictions in the results of XRD data interpretation in [19,20]. It turned out that the space group and the parameters of the crystal cell in [20] coincide with the data of [19]. However, in [20] it was shown that the unit cell contains four octanuclear complexes of composition Co_8_(µ_4_–O)_2_(µ_2_–OOCCMe_3_)_6_(µ_3_–OOCCMe_3_)_6_ [hereinafter Co_8_O_2_(piv)_12_] rather than eight tetranuclear complexes Co_4_O(piv)_6_, as in [19].

In the study [21], it was demonstrated that thermal decomposition of various Co(II) carboxylate complexes in the temperature range of 190–220 °C is accompanied by aggregation to form the octanuclear complex Co_8_O_2_(piv)_12_. This complex was obtained as the major product of the solid state decomposition of polymeric cobalt carboxylate [Co(piv)_2_]_n_ (below 620 K) [21]. The authors [21] presumed that the octanuclear complex itself is sublimed without decomposition, and the evaporation intensity sharply increases above 500 K.

Another conclusion about the composition of cobalt complexes in the gas phase over cobalt(II) oxotrimethylacetates was made in [22], where the processes of vaporization of cobalt(II) pivalate, and cobalt(II) oxopivalate were studied. In the mass spectrum of saturated vapor over cobalt(II) oxopivalate in the temperature range 403–448 K, ions with one molecular precursor, the tetranuclear complex Co_4_O(piv)_6_, were registered.

Thus, in [19,20,21,22], there are contradictions either about the composition and structure of polynuclear complexes of cobalt(II) oxopivalate in the crystalline state (the tetranuclear complex in [19] versus the octanuclear complex in [20]), or in the gas phase (the tetranuclear complex in [22] and against the octanuclear complex in [21]).

In addition, the molecules Co_4_O(piv)_6_ and Co_8_O_2_(piv)_12_ present the compounds possessing outstanding spin features. The complex Co_4_O(piv)_6_, as well as Co_8_O_2_(piv)_12_, can have different spin states, among which the states with a maximum possible spin equal to 6 and 12 correspondently.

The purpose of this work is to determine the composition and geometric structure of the cobalt(II) oxopivalate complex in the gas phase, as well as to determine the energies of the complex in electronic states with different multiplicities by quantum chemistry methods, and to compare the structure of complexes existing in the condensed and gas phases. For this purpose, a synchronous gas electron diffraction/mass spectrometry experiment (GED/MS) was performed, and a series of quantum chemical calculations were carried out.

## 2. Results and Discussions

### 2.1. Vapor Composition of Cobalt(II) Oxopivalate at T = 410 K

The mass spectrum recorded at T = 410 K simultaneously with the registration of diffraction pattern is shown in Table 1. It is interesting that low-intensity [Co_8_O_2_(piv)_9_O]^+^, [Co_8_O_2_(piv)_8_O_2_]^+^, [Co_6_O_2_(piv)_7_O_2_]^+^ ions were detected in the mass spectrum. Indirectly, they may indicate that the crystalline phase of cobalt(II) oxopivalate consists of octanuclear complexes Co_8_O_2_(piv)_12_, as is shown in [20]. However, the [Co_4_O(piv)_6_]^+^, [Co_4_O(piv)_5_]^+^, [Co_4_O(piv)_3_-tb]^+^ ions have many times greater intensity in the mass spectrum of saturated vapor. Their elemental compositions correspond to the dominance of tetranuclear complexes Co_4_O(piv)_6_ in the gas phase.

This result coincides with the conclusion of the authors of [22], who reported on cobalt(II) oxopivalate vapors consisting of tetranuclear complexes. The authors of [22] studied the mass spectra of the gas phase of this compound at different energies of ionizing electrons and at a close temperature (T = 430 K). As the ionizing voltage decreases from 60 to 14.2 V, the intensity of the [Co_4_O(piv)_6_]^+^ parent ion increases, and this ion remains the only one in the mass spectrum at lowest U_ioniz_.

The strong dominance of the tetranuclear complex Co_4_O(piv)_6_ in the saturated vapor of cobalt(II) oxopivalate makes it possible to interpret the electron diffraction data based on the assumption that a single molecular form is present in the scattering volume.

### 2.2. Electronic States of the Tetranuclear Co(II) Complex with Different Multiplicity. Results of Quantum Chemical Calculations

In the structural analysis of electron diffraction data, the initial geometric model of the molecule under study is usually estimated using quantum chemical calculations and then refined using experimental data. The case with the Co_4_O(piv)_6_ complex is complicated by the fact that the complex can exist in electronic states of different multiplicities. In such cases, the geometric structure can either change significantly with a change in multiplicity, as, for example, in Ni(acacen) [23] and Ni(salen) [24] complexes, or these changes turn out to be insignificant, as, for example, in complexes (CrCl_2_)_n_ [25]. It was shown in [23,24] that the spin state affects significantly the geometric structure of the Ni(salen) and Ni(acacen) molecules. For both complexes in the low-spin state, the coordination cavity has a planar structure. In the case of a high-spin state, the structure of the coordination cavity is a distorted tetrahedron with elongated Ni–N and Ni–O distances.

However, the dimer and the tetramer complexes (CrCl_2_)_n_ in the different spin states differ only little in energy and have very similar geometries [25].

The presence of four cobalt atoms bonded through oxygen atoms and possessing a partially filled 3d shell suggests that the Co_4_O(piv)_6_ complex has several electronic states with different multiplicities. However, an attempt at a practical application of theoretical approximations focused on the study of the electronic structure of the large Co_4_O(piv)_6_ complex has led to the need to use computational resources that are beyond our capabilities. Therefore, when studying the electronic structure of the tetranuclear complex, the oxopivalate group in Co_4_O(piv)_6_ was replaced by the formate group HCO_2_, and the study of the electronic structure was carried out using the example of cobalt oxoformate Co_4_O(formyl)_6_, which has the same skeleton Co_4_(µ_4_-O)(µ_2_-OOC)_6_ (calculation details are given in Section 3.2).

Full optimization of the Co_4_O(formyl)_6_ complex was performed in the DFT/PBE0 approximation for the high-spin state (M = 13). The equilibrium geometry had T_d_ symmetry (see Figure 1). The following values of geometric parameters were obtained: r_e_(Co–O_c_) = 1.977 Å, r_e_(Co–O_L_) = 1.961 Å, r_e_(C–O_L_) = 1.250 Å, r_e_(C–H) = 1.100 Å, O_L_-Co-O_c_ = 110.8°, O_L_-C-O_L_ = 127.8°.

Within the framework of the crystal field theory, each cobalt cation Co^2+^(3d^7^) is located inside a trigonal pyramid formed by four oxygen atoms. The pyramid is close in shape to a tetrahedron and has local C_3v_ symmetry. The splitting of a spherical 3d shell in crystal fields of T_d_ and C_3v_ symmetry, the population of 3d orbitals with seven electrons with the formation of a high-spin electronic state of Co, and the correlation with high-lying molecular orbitals of the Co_4_O(formyl)_6_ complex are shown in Figure 2.

The analysis of molecular orbital composition for the high-spin state showed that each of the 20 highest-occupied MOs corresponds to different combinations of the d-orbitals of four cobalt atoms. Eight of them are doubly occupied (DOMO), and 12 are singly occupied orbitals (SOMO). Obviously, eight doubly occupied and 12 singly occupied molecular orbitals can be formed by various combinations of doubly occupied *e* and singly occupied *t*_2_ or *e* + *a*_1_ 3d orbitals of four cobalt atoms (Figure 2). This assumption is confirmed by the fact that the energy gap between doubly and singly occupied MOs is ~1 eV. At the same time, the energy differences within a set of eight DOMOs and a set of 12 SOMOs do not exceed 1.1 and 0.3 eV, respectively (Figure 2).

According to the diagram in Figure 2, the active space of the CASSCF method included 12 electrons and 12 molecular orbitals (SOMO). The choice of such an active space makes it possible to perform calculations for states with multiplicities M = 1, 3, 5, 7, 9, 11, and 13. The relative energies ΔE of states with different multiplicities were obtained with the values of geometric parameters fixed at values optimized in approximation DFT/PBE0 for the high-spin electronic state (see above).

Table 2 and Figure 3 present the results of calculating the relative energies of states with different multiplicities in the CASSCF and XMCQDPT2 approximations.

According to Table 2 and Figure 3, considering the electron correlation leads to greater energy stabilization of low-spin electronic states. As a result, the relative energy of the high-spin electronic state (M = 13) increases by more than four times and reaches 6.48 kJ/mol.

The forms of active MO are shown in Figure 4.

An analysis of the obtained data shows that for the ground electronic states of each considered multiplicity, the forms of active MOs remain practically unchanged. The shape of these MOs (Figure 4) confirms the above conclusion that each active MO corresponds to different combinations of the components of the 3d orbitals of four cobalt atoms. Moreover, judging by Figure 4, the interaction of the 3d–AO components belonging to different cobalt atoms is practically absent.

The wave function of the CASSCF method is a series expansion in terms of the configuration state functions (CSFs). Each CSF can be represented as one of the options for distributing electrons over a set of active orbitals while maintaining a specific multiplicity of the electronic state. The coefficients, with which CSFs enter the wave function, as well as the composition of the populated MOs, vary in the iterative procedure. Table 2 shows the averaged populations (Q_av_) of active MOs for the ground electronic states of each considered multiplicity. The Q_av_ values are obtained by summing the populations (0, 1, or 2) of a particular MO in each CSF participating in the approximation of the wave function of the considered electronic state, considering the weight of this CSF. The populations Q_av_ of all MOs in all considered electronic states are practically equal to 1 (Table 2). These data, together with the results of the CASSCF calculation, show that it is impossible to single out any CSF with a dominant weight in the composition of the wave functions. The values of the coefficients with which CSF enter the wave functions of electronic states with M = 1–11 do not exceed 0.05.

To determine the magnitude of the change in the geometric parameters of the coordination center [Co_4_(µ_4_-O)(µ_2_-OOC)_6_] with a change in the multiplicity of the electronic state (M = 13, 7 and 1), two-parameter optimization (distances r(Co-O_c_) and r(Co-O_L_)) by the numerical energy method [26,27] in the MCQDPT2 approximation was used.

The C–O_L_, C–H bond lengths and bond angle values were fixed at the values optimized in the DFT/PBE0 (see above). Optimization led to the following bond lengths (Å) and relative energies (ΔE, kJ/mol): M = 13 − r_e_(Co–O_c_) = 1.966, r_e_(Co–O_L_) = 1.951, ΔE = 7.59; M = 7 − r_e_(Co–O_c_) = 1.961, r_e_(Co–O_L_) = 1.950, ΔE = 2.38; and M = 1 − r_e_(Co–O_c_)= 1.959, r_e_(Co–O_L_) = 1.951, ΔE = 0. The most important conclusion is that the transition from high-spin to low-spin electronic states leads to a systematic decrease in the Co–O_c_ bond length. However, this change is small and does not exceed 0.007 Å, which is comparable with the error limit in determining this parameter in an electron-diffraction experiment. The Co–O_L_ bond length remains practically unchanged. The singlet term has the lowest energy, and the relative energy of the electronic state with M = 13 increases with geometry relaxation to 7.59 kJ/mol.

As noted above, a similar result was obtained in [25] for the dimer Cr_2_Cl_4_. Each Cr atom in dimer has four uncoupled electrons. The authors of this work note that the difference in energy between states with ferromagnetically and antiferromagnetically coupled Cr atoms is small, and this circumstance indicates little spin-coupling between metal atoms in the dimer. As an example, they consider nonet and singlet states of dimers, which differ in energy only by 2.3 kJ/mol (0.024 eV). In such situations, the high-spin state should be realized in accordance with Hund’s rule. However, the calculations predict that both in the case of Co_4_O(formyl)_6_ and in the case of Cr_2_Cl_4_ complexes, the low-spin singlet state has a lower energy compared to high-spin ones.

### 2.3. Geometric Structure of the Co_4_O(piv)_6_ in the Gas Phase. Results of Structural Analysis of Electron Diffraction Data

According to mass-spectra, the saturated vapor of cobalt(II) oxopivalate complex at the temperature of GED/MS experiment contains the single molecular form Co_4_O(piv)_6_ practically. The heavier molecular form Co_8_O_2_(piv)_12_ is present at a level of 1–2% and can be neglected in the structural analysis.

The Co_4_O(piv)_6_ compound differs from Co_4_O(formyl)_6_ considered in Section 2.2 only by the presence of tert–butyl substituents in the ligands, which cannot significantly affect the electronic states of Co atoms and the difference in the energy of states with different multiplicities.

In this case, at the temperature of the GED experiment, the thermal energy RT exceeds the difference in the energies of seven electronic states (Table 2, CASSCF). Therefore, a mixture of Co_4_O(piv)_6_ complexes with different multiplicities can exist in saturated vapor. Moreover, the proximity of the geometric structure of these complexes in different electronic states, which is due to the similar spatial shape of active MOs (Figure 4), allows us to use the structure calculated by the DFT B3LYP/cc-pVTZ method for a state with a multiplicity of 13 as a starting geometric model. The atomic coordinates of the optimized structure and the vibration frequencies are provided in the Supporting Information file (Appendix A). No imaginary frequencies confirm that the optimized geometry corresponds to the true minimum.

Details of structural analysis of GED data are described in Section 3.3.

The experimental values of the geometric parameters of the Co_4_O(piv)_6_ complex of C_3_ symmetry in the gaseous state (Figure 5) are provided in Table 3. The values of the parameters Co_4_O(piv)_6_ calculated by the DFT method, as well as the corresponding parameters of the Co_8_O_2_(piv)_12_ [20] and Co_4_O(OOCNC_9_H_18_)_2_ [28] complexes in the crystal, are provided for comparison.

The functions of the molecular-scattering intensity sM(s) and the radial distribution f(r) are shown in Figure 6 and Figure 7.

It can be seen that the experimental and calculated geometric parameters of the free Co_4_O(piv)_6_ molecule generally agree within the experimental error. At the same time, calculations with the B3LYP functional better predict the parameters of the ligand, while the PBE0 functional better conveys the structure of the coordination center. The values of the Co–O bond lengths and the O–Co–O and Co–O–Co bond angles in the central OCo_4_ fragment, and four O_c_CoO_3_ fragments indicate that the structure of these fragments is close to tetrahedral. The large uncertainty in the value of the O_L_–C–C_t_–C_m_ torsion angle, which determines the position of the tert–butyl substituent (tb), relative to the backbone of the complex, may indicate a large amplitude of hindered internal rotation of the tb groups at the temperature of the GED experiment. The agreement of the experimental functions sM(s) and f(r) with their theoretical analogs (Figure 6 and Figure 7) indicates the validity of the assumption about the dominance of the tetranuclear complex of C_3_ symmetry in the saturated vapor of cobalt(II) oxopivalate.

### 2.4. Nature of Chemical Bonds in the Co_4_O(formyl)_6_ and Co_4_O(piv)_6_ Complexes

For the Co_4_O(formyl)_6_ and Co_4_O(piv)_6_ possessing the same cores, the electron density distribution for high-spin states (M = 13) was analyzed using the QTAIM [29,30] and NBO [31] methods, respectively. The results of the analysis are shown in Table 4.

Note that for the Co–O bonds, the QTAIM and NPA [31] data do not contradict each other. An effective positive charge on Co atoms differs markedly from +2. Accordingly, the oxygen atoms O_c_ and O_L_ bear negative charges, and according to the QTAIM data, their difference is small, while according to the NPA data, the difference in charges is very significant.

The q, *ρ,* δ (QTAIM), and q(NPA) values indicate that the Co–O_c_ and Co–O_L_ bonds can be attributed to ionic (or coordination) bonds (∇^2^*ρ* ˃ 0) [29] with a significant contribution of the covalent component (δ ˃ 0 and P ˃ 0). The C–O_L_ bond, despite the large difference in the charges on the C and O_L_ atoms, and according to the values of ∇^2^*ρ* ˂ 0, delocalization index δ, and bond order P, can be attributed to the category of a polar covalent bond, which is much stronger than the Co–O bonds.

Natural electron configurations of atoms in molecules show that in the Co_4_O(piv)_6_ complex, there is a significant transfer of electron density from the valence 4s orbitals of the cobalt atom to the 2p AO of oxygen. As a result of reverse donation, a partial transfer of electron density from oxygen atoms to 4p–AO of Co atoms occurs, which reduces the positive charge on them (+1.2 instead of +2).

### 2.5. Structure of Cobalt Carboxylates in the Gas and Crystalline Phases

As noted in the introduction, the crystalline phase of cobalt(II) oxopivalate consists of octanuclear complexes Co_8_O_2_(piv)_12_ [20] with two tetra-coordinated and six five-coordinated Co atoms. The CoO_4_ fragments have a structure close to tetrahedral (the geometrical parameters are provided in Table 3), while the CoO_5_ fragments have the structure of a distorted trigonal bipyramid in which the axial Co–O distances (2.093 and 2.212(3) Å) are significantly longer than the equator ones (r(Co–O) = 1.966, 1.974 and 2.020 Å). As a result, two OCo_4_ (piv)_6_ fragments can be distinguished in the structure of the Co_8_O_2_(piv)_12_ complex, which are linked by six long Co–O bonds (Figure 8a). Apparently, these bonds are broken during the sublimation of the sample.

It is of interest to compare the structure of the backbones in the tetranuclear cobalt complexes [Co_4_(µ_4_-O)(µ_2_-OOC-R)_6_], where R= -H, -C(CH_3_)_3_ and -NC_9_H_18_ (2,2,6,6-tetramethylpiperidin) with different R substituents in the chelate ligand (Table 3). X-ray diffraction analysis [28] shows that, in the latter case, the unit cell (space group P2_1_/n) contains eight complexes [Co_4_(µ_4_–O)(µ_2_–OOCNC_9_H_18_)_6_]. Moreover, the geometric parameters of the [Co_4_(µ_4_–O)(µ_2_–OOC)_6_] core of the gaseous Co_4_O(piv)_6_ complex studied in this work are close to the parameters of the [Co_4_(µ_4_–O)(µ_2_–OOCNC_9_H_18_)_6_] complex core in the crystal. This indicates a weak effect of the nature of the substituent R on the backbone of the complexes. Therefore, the conclusions about the close energy of electronic states with different multiplicities, made in Section 2.2, can be valid for all [Co_4_(µ_4_–O)(µ_2_–OOC-R)_6_] complexes with a different nature of substituents R.

## 3. Materials and Methods

### 3.1. Conditions of the Synchronous GED/MS Experiments

The combined EMR-100/APDM-1 device [32,33] was used for collecting the diffraction patterns simultaneously with recording the mass spectra (GED/MS experiment). To determine the wavelength of fast electrons, diffraction patterns of polycrystalline zinc oxide were collected before and after the diffraction experiment for Co_4_O(piv)_6_ vapor. The effusion cell made of stainless steel X18H10T was used for vaporization of the sample. The temperature of the sample during GED/MS experiment was measured by a thermocouple W-Re-5/20. Main conditions of the GED/MS experiment are listed in Table 5.

To measure the optical densities of diffraction patterns, a modified microdensitometer MD-100 was used [34]. Experimental molecular scattering intensity were calculated as sM(s) = [I_exp_(s) − G(s)]∙s/G(s), where *I*_exp_(*s*) is a total scattering intensity function and G(s) is an empirical background function.

### 3.2. Details of Quantum Chemical Calculation of Co_4_O(formyl)_6_ in Different Electronic States

Theoretical calculations for cobalt oxoformate Co_4_O(formyl)_6_, where formyl is the formate group (HCO_2_), were carried out using the DFT/PBE0 [35] and CASSCF approximations, followed by the consideration of the electron correlation in the framework of the multiconfiguration quasi-degenerate perturbation theory of the second order (XMCQDPT2) [36,37]. The three-exponential Sapporo basis set (SPK-TZC) was used for all atoms: (18s13p103f2g/8s6p4d2f1g)–Co [38]; (10s6p3d2f/6s4p2d1f)–C, O [38]; (6s3p2d/3s2p1d)–H [39]. All calculations were performed using the FireFly 8.2.0 program [40].

For XMCQDPT2 calculations, the doubly occupied molecular orbitals corresponding to 1s orbitals of C and O, and 1s, 2s, 2p orbitals of Co, were included in the frozen core.

The intruder state avoidance method (ISA) was applied to solve the problem of small-energy denominators, which can appear in the perturbative expansion [41]. The value 0.02 was chosen for parameter controlling the energy-denominator shift [37].

The reason for using the PBE0 functional is its good performance in predicting molecular geometries of transition metal complexes [42,43,44,45,46]. The choice of the CASSCF and XMCQDPT2 approximations is due to the expected multireference nature of the wave functions for a number of considered electronic states.

The topological analysis of electron density distribution function was carried out using AIMAll professional software [30].

### 3.3. Features of Structural Analysis of GED/MS Data

According to mass spectra, the saturated vapor of cobalt(II) oxopivalate at the temperature of GED/MS experiment contains the single molecular form Co_4_O(piv)_6_.

In the least squares analysis of electron diffraction data, modified KCED program [47] was used.

The geometry configuration of the complex was described by 12 independent parameters. The set of independent parameters included: four bond lengths Co–O_c_, C–O_L_, C–C_t_, C–H, six bond angles Co1–O_c_–Co2, O_L_–Co1–O_c_, O_L_–C–C_t_, C–C_t_–C_m_, C_m_–C_t_–C_m_, C_t_–C_m_–H, two φ_1_(O_L_–C–C_t_–C_m_), and φ_2_(H–C_m_–C_t_–C) torsion angles, which take into account the turning of the tert–butyl and methyl groups. The parameters obtained from the results of quantum chemical calculations (B3LYP/cc-pVTZ) were obtained as the starting values of internuclear distances, bonds, and torsion angles of Co_4_O(piv)_6_ complex (M = 13).

The geometry of the molecule was investigated in terms of the r_h1_ structure. The vibrational corrections to the internuclear distances Δ*r* = *r_a_* − *r*_*h*1_ and the starting root–mean–square vibration amplitudes needed for the LS analysis of molecular scattering intensity were calculated in harmonic approximation by the 0VibModule program [48] considering the non-linear interrelation between internal and Cartesian vibrational coordinates. The quantum–chemically calculated difference between parameters of the same type was retained. To decrease the correlation between varied during LS analysis parameters, vibrational amplitudes for atom pairs possessing the close internuclear distances were refined in groups corresponding to the peaks on the radial distribution curve *f*(*r*) (Figure 7). In the least squares analysis, both 12 geometrical parameters (Table 3) along with 21 groups of vibration amplitudes (Appendix A, Appendix A) were varied simultaneously.

## 4. Conclusions

In the range of 30–1800 Da, the mass spectrum of the oxopivalate complex of cobalt(II) was registered. It has been found that at T = 410 K, tetranuclear Co_4_O(piv)_6_ complex dominates in the gas phase instead of eight nucleus Co_8_O_2_(piv)_12_ complexes in the crystal.

The presence of an open 3d shell for each Co atom leads to the possibility of existence of electronic states with different multiplicities for the Co_4_O(piv)_6_ complex. For Co_4_O(formyl)_6_, which has the same core as Co_4_O(piv)_6_, the relative energies of states with Multiplicities 1, 3, 5, 7, 9, 11, and 13 were calculated in the CASSCF and XMCQDPT2 approximations. It is shown that the states have similar energies and identical shapes of active orbitals. Each active MO corresponds to different combinations of the 3d orbitals of four cobalt atoms.

The interaction of the 3d–AO belonging to different cobalt atoms is practically absent. As a result, the geometric structure of the tetranuclear complex remains virtually unchanged in electronic states of different multiplicities. There is only a very small change in energy from the antiferromagnetically to the ferromagnetically coupled Co atoms.

The structure of the free molecule Co_4_O(piv)_6_ was studied for the first time by the gas electron-diffraction method. It was found that the compound is thermally stable at least to the temperature 410 K of the electron-diffraction experiment.

The geometry of the Co_4_O(piv)_6_ complex has C_3_ symmetry; the values of the Co–O bond lengths and the O–Co–O and Co–O–Co bond angles in the central OCo_4_ fragment and four CoO_4_ fragments indicate that these fragments have the structure of distorted tetrahedra.

Using the QTAIM and NBO methods for the Co_4_O(formyl)_6_ and Co_4_O(piv)_6_ complexes, having the same backbones, it was shown that the Co–O_c_ and Co–O_L_ bonds can be attributed to ionic (or coordination) bonds with a significant contribution of the covalent component.

The closeness of the geometric and electronic characteristics of the backbone of tetranuclear complexes [Co_4_(μ_4_–O)(OOC–R)_6_], where R = -H, -C(CH_3_)_3_, and -NC_9_H_18_, which suggests that, regardless of the nature of the substituent in such complexes, a transition from antiferromagnetic to ferromagnetic states is possible with a small change in energy.

The results of the performed studies make it possible to recommend cobalt(II) oxopivalate as precursors in the preparation of oxide films or coatings in the CVD technique, since these compounds are characterized by high volatility, a simple composition of the gas phase, congruent nature of vaporization, and constancy of thermodynamic properties over a long period of sublimation.

## Figures and Tables

**Figure 1 ijms-24-13224-f001:**
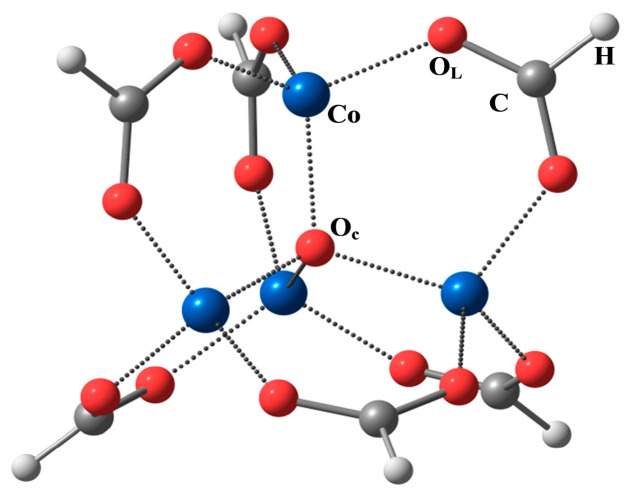
Structure of the Co_4_O(formyl)_6_ molecule (T_d_ symmetry) and atomic designations.

**Figure 2 ijms-24-13224-f002:**
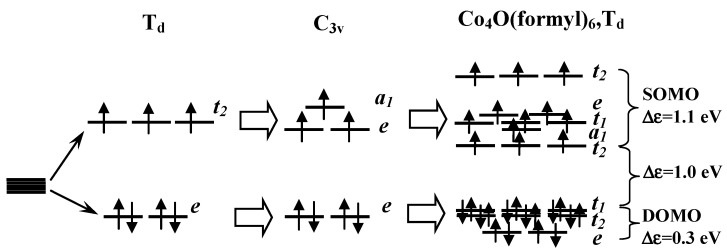
Splitting of five-fold degenerate 3d orbitals of cobalt in a crystal field of T_d_ and C_3v_ symmetry; symmetry and occupation of 3d orbitals in the high-spin electronic state; correlation with molecular orbitals of the Co_4_O(formyl)_6_ complex; Δε—energy differences of molecular orbitals.

**Figure 3 ijms-24-13224-f003:**
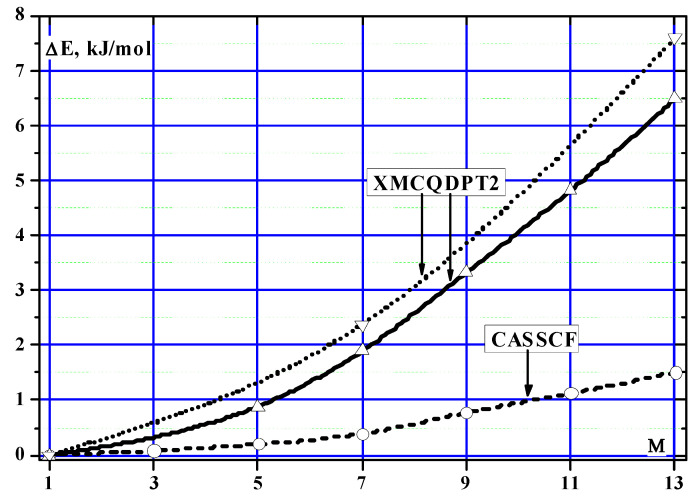
Correlation of the relative energy (ΔE) of the spin state of the Co_4_O(formyl)_6_ complex with the multiplicity of this state from the CASSCF and XMCQDPT2 calculations. Solid XMCQDPT2 line–calculation with fixed geometric parameters. Dashed XMCQDPT2 line corresponds to calculation with optimized Co–O distances.

**Figure 4 ijms-24-13224-f004:**
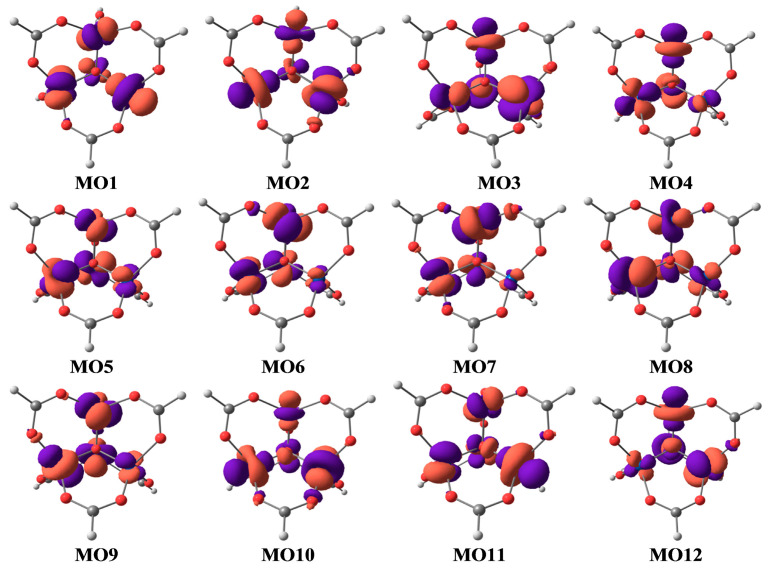
Forms of active MOs.

**Figure 5 ijms-24-13224-f005:**
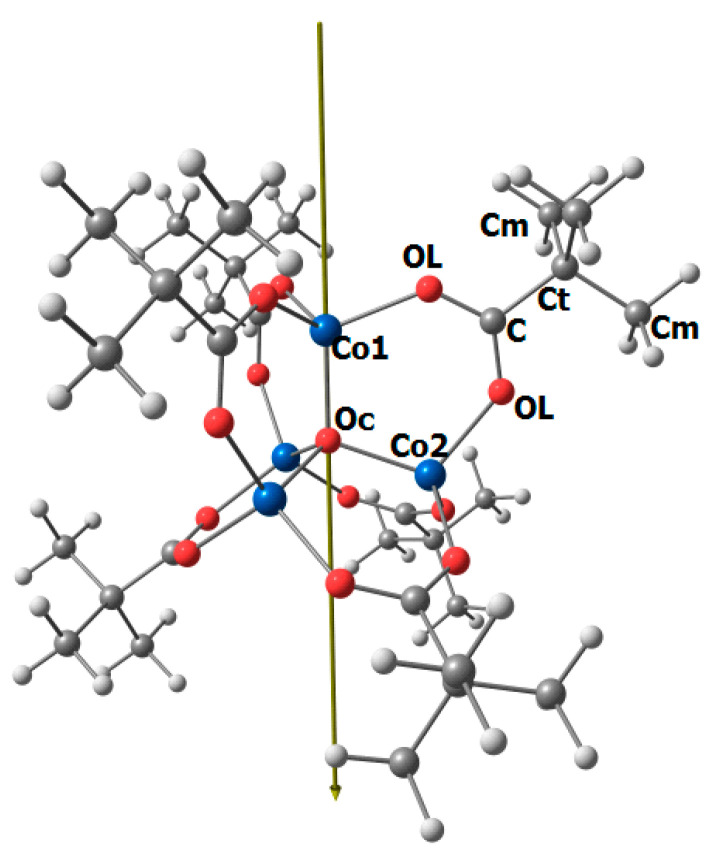
Geometric structure of the Co_4_O(piv)_6_ complex of C_3_ symmetry with atom numbering.

**Figure 6 ijms-24-13224-f006:**
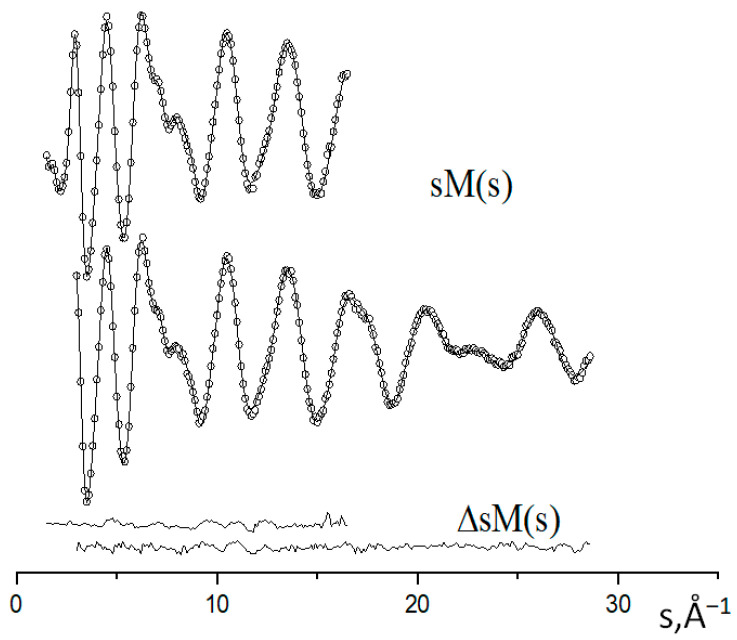
Experimental (dots) and calculated (solid line) molecular intensities sM(s) and their differences ΔsM(s) for the long (a) and short (b) nozzle-to-plate distances.

**Figure 7 ijms-24-13224-f007:**
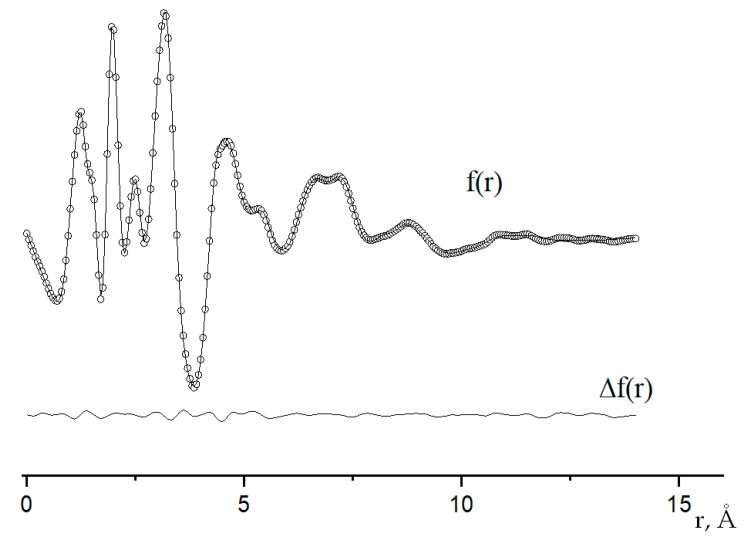
Experimental (dots) and theoretical (solid line) radial distribution curves f(r) and their differences Δf(r).

**Figure 8 ijms-24-13224-f008:**
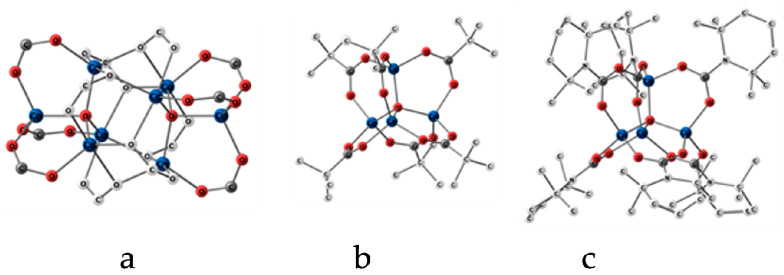
Structure of cobalt(II) oxopivalate [Co_8_O_2_(piv)_12_] complex in crystal (**a**), without tb-groups, [OCo_4_(piv)_6_] complex in gas (**b**), [Co_4_O(OOCNC_9_H_18_)_6_] complex in crystal (**c**). Hydrogen atoms are not shown.

**Table 1 ijms-24-13224-t001:** Relative intensity of ion currents and elemental composition of ions recorded in the mass spectra of cobalt(II) oxopivalate.

m/e	Ion	I_rel_, U_ioniz_ = 50 V T = 410 K	I_rel_ [22] U_ioniz_ = 60 V T = 430 K
1429	[Co_8_O_2_(piv)_9_O]^+^	2	
1344	[Co_8_O_2_(piv)_8_O_2_]^+^	5	
1093	[Co_6_O_2_(piv)_7_O_2_]^+^	1	
858	[Co_4_O(piv)_6_]^+^	13	10.2
757	[Co_4_O(piv)_5_]^+^	100	100
656	[Co_4_O(piv)_4_]^+^	1	11.3
571	[Co_4_O(piv)_3_O]^+^	1	
555	[Co_4_O(piv)_3_]^+^	1	12.1
498	[Co_4_O(piv)_3_-tb]^+^	25	100.6
395	[Co_3_O(piv)_2_]^+^	1	
59	[Co]^+^	5	20.7

**Table 2 ijms-24-13224-t002:** Relative energies (ΔE, kJ/mol) and average occupations (Q_av_, e) of 12 active MOs for electronic states of various multiplicities (M = 2S + 1).

	Spin State ^1^						
Multiplicity, M	13	11	9	7	5	3	1
CASSCF, ΔE	1.51	1.12	0.77	0.39	0.20	0.08	0.00
XMCQDPT2, ΔE	6.48	4.82	3.32	1.90	0.89	-	0.00
Q_av_(MO1)	1.000	1.007	1.013	1.029	1.024	1.025	1.026
Q_av_(MO2)	1.000	1.007	1.013	1.020	1.020	1.024	1.026
Q_av_(MO3)	1.000	1.007	1.013	1.021	1.026	1.025	1.026
Q_av_(MO4)	1.000	1.003	1.005	1.005	1.012	1.010	1.010
Q_av_(MO5)	1.000	1.000	1.000	1.000	1.000	1.000	1.000
Q_av_(MO6)	1.000	1.000	1.000	1.000	1.000	1.000	1.000
Q_av_(MO7)	1.000	1.000	1.000	1.000	1.001	1.000	1.000
Q_av_(MO8)	1.000	0.999	0.999	0.998	0.995	0.998	0.998
Q_av_(MO9)	1.000	0.999	0.999	0.999	0.999	0.998	0.998
Q_av_(MO10)	1.000	0.992	0.985	0.979	0.977	0.973	0.972
Q_av_(MO11)	1.000	0.992	0.985	0.970	0.975	0.973	0.972
Q_av_(MO12)	1.000	0.992	0.985	0.979	0.972	0.973	0.972

^1^ Numbers of configuration state functions (CSF) used to approximate the wave function of each considered electronic state with specific multiplicity (M) are 1 (M = 13), 143 (M = 11), 4212 (M = 9), 44,044 (M = 7), 196,625 (M = 5), 382,239 (M = 3), and 226,512 (M = 1).

**Table 3 ijms-24-13224-t003:** Selected structural parameters of gaseous Co_4_O(piv)_6_ complex and Co_8_O_2_(piv)_12_ and Co_4_O(OOCNC_9_H_18_)_2_ complex in the crystalline state.

Complex, Parameter ^a^	Co_4_O(piv)_6_		Co_8_O_2_(piv)_12_	Co_4_O(OOCNC_9_H_18_)_2_
Method	GED/MS 410 K r_h1_ (C_3_) R_f_ = 4.5%	B3LYP r_e_ (C_3_)	X-ray ^b^ [20]	X-ray [28]
r(O_C_–Co1) ^c^	1.975 (5) ^d^ *p*_1_ ^e^	1.987	1.984	1.955
r(O_C_–Co2)	1.972 (*p*_1_)	1.984	1.966	1.936, 1.941, 1.954
r(Co1–O_L_) ^c^	1.963 (5) (*p*_1_)	1.973	1.942	1.936, 1.939, 1.942
r(Co2–O_L_)	1.965 (*p*_1_)	1.975	-	1.919–1.950
r(O_L_–C)	1.267/1.265(4) *p*_2_	1.265/1.263	1.247/1.257	1.240–1.288
r(C-Ct)	1.537(4) *p*_3_	1.537	1.519	-
r(C_t_–C_m_)	1.532 (*p*_3_)	1.532	1.514	-
r(C–H)	1.079(5) *p*_4_	1.09	(0.96)	
Co1–O_C_–Co2	110.4(6) *p*_5_	109.5	101.4	108.3–109.5
O_L_–Co1–O_C_	111.8(3) *p*_6_	110.4	113.7	110.2–111.9
O_L_–Co1–O_L_	107.1(3) (*p*_6_)	108.5	104.9	107.5
Co1–O_L_–C	131.8(6) (*p*_5_)	132.5	132.7	131.9–133.2
Co2–O_L_–C	128.8(6) (*p*_5_)	130.7	-	131.6–132.3
O_L_-C–Ct	117.2(13) *p*_7_	117.0	119.3	-
C–C_t_–C_m_	112.1(12) *p*_8_	111.2	112.1	-
C_m_–C_t_–C_m_	108.3(27) *p*_9_	110.0	110.3	-
C_t_–C_m_–H	111.0(18) *p*_10_	109.6	(109.5)	-
O_L_–C–C_t_–C_m_	5.3(119) *p*_11_	0.1	11.9	-
H–C_m_–C_t_–C	178.5(45) *p*_12_	180	177.3	-

^a^ Å, deg. ^b^ Parameters related to O_c_Co1(O_L_)_3_ fragments with tetracoordinated Co1 atoms are provided. ^c^ O_c_—central oxygen atom; O_L_—bridging oxygen atom. ^d^—noted in parentheses values are the error limits for distances, calculated as σ = (σ_scale_^2^ + (2.5σ_LS_)^2^)^1/2^, where the scale error σ_scale_ = 0.002r; the error in determining bond angles was provided equal to 3σ_LS_, and the error in torsion angles was 2.5σ_LS_. ^e^—*p_i_*—independently refined parameter; (*p_i_*)—parameter refined in the i-th group.

**Table 4 ijms-24-13224-t004:** Net atomic charges (q, e) and topological parameters * (*ρ*, ∇^2^*ρ*, ε, δ) of the electron density distribution in bond critical points (BCP) of the complex Co_4_O(formyl)_6_. Net atomic charges (qNPA) and bond order P by Wiberg [31] into complex Co_4_O(piv)_6_.

	QTAIM Co_4_O(formyl)_6_	NBO Co_4_O(piv)_6_		BCP, QTAIM Co_4_O(formyl)_6_				NBO Co_4_O(piv)_6_
Atom	q	q_NPA_	Bond	*ρ*	∇^2^*ρ*	ε	δ	P
Co	1.41	1.20	Co-O_c_	0.087	+0.425	0.00	0.52	0.28
O_c_	−1.32	−1.20	Co-O_L_	0.087	+0.456	0.01	0.47	0.30
O_L_	−1.24	−0.72	C-O_L_	0.383	−0.592	0.07	1.06	1.37
C	1.74	0.84						
Natural electron configuration (NBO, Co_4_O(piv)_6_)								
O_c_	[core] 2s(1.75) 2p(5.43)							
Co	[core] 4s(0.22) 3d(7.25) 4p(0.32)							
O_L_	[core] 2s(1.67) 2p(5.04)							

* *ρ*–electronic density at BCP, e/Bohr ^3^; ∇^2^*ρ*—the Laplacian of ρ(r) (∇^2^*ρ* = λ_1_ + λ_2_ + λ_3_, λ_1_, λ_2_, λ_3_—the eigenvalues of the Hessian matrix of (r)), e/Bohr ^5^; ε = λ_1_/λ_2_ − 1—a bond ellipticity; δ–a electron delocalization index, e.

**Table 5 ijms-24-13224-t005:** Conditions of the synchronous GED/MS experiments.

Nozzle-To-Plate Distance, mm	338	598
Electron beam current, µA	1.4	1.0
Temperature of effusion cell, K	410 (5)	411 (5)
Accelerating voltage, kV	83.5	81.4
Ionization voltage, V	50	50
Exposure time, s	100	45
Residual gas pressure, Torr	1.4 × 10^−6^	2.0 × 10^−6^
Scattering angles, Å^−1^	3.0–28.6	1.5–16.5

## Data Availability

No new data were created.

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
