# Peer review of "Molecular Structure of Gaseous Oxopivalate Co(II): Electronic States of Various Multiplicities"

_ijms, 2023, doi:10.3390/ijms241713224_

Round 1

Reviewer 1 Report

In this manuscript, the authors studied the composition and structure of molecular forms existing in a saturated vapor of cobalt(II) oxopivalate at Т=410 K. For the purpose of the study, the authors used synchronous electron diffraction/mass spectrometry technique and ab initio calculations. The results of the vapor composition analysis of cobalt(II) oxopivalate at T=410 K are in agreement with the conclusions of previous studies, which showed that cobalt(II) oxopivalate vapors consist of tetranuclear complexes. Additionally, the results of the ab initio calculations showed that the Co-Oc and Co-O bonds can be attributed to ionic or coordination bonds with a significant contribution of the covalent component. I believe that this is overall a useful paper with interesting results. Overall, I find this paper to be useful with interesting results. However, there are some minor issues that need to be addressed before publication.

1. The introduction section should be extended to provide a more detailed explanation of the significance of both the CVD technique and the cobalt pivalate complexes as potential precursors.

2. The authors should clarify the reason for selecting this specific level of theory, especially the PBE0 functional, for geometry optimizations. Adequate justification, including references to fundamental reasons or previous studies (including benchmark studies), supporting this choice of DFT functional, should be provided and explained in the text.

3. It would be beneficial if the authors performed vibrational frequency calculations to confirm that the optimized geometries correspond to the true minima.

4. The Cartesian coordinates of the optimized structure must be included in the Supporting Information file.

Upon addressing these points, I recommend accepting the manuscript for publication.

Author Response

Dear Reviewer,

The authors of this manuscript are deeply grateful to you for the comments, which motivate us really improve the content and representation of manuscript. We tried to follow all your suggestions.

Please find below our responds.

Comment 1

The introduction section should be extended to provide a more detailed explanation of the significance of both the CVD technique and the cobalt pivalate complexes as potential precursors.

Response

In accordance with this comment, we have supplemented the Introduction section with additional information about the features of the application of the CVD method and more detailed information about the products obtained by CVD using organometallic complexes as precursors. Relevant references have been added to the list of cited literature. Additions to the text and in the References are in red font.

Comment 2

The authors should clarify the reason for selecting this specific level of theory, especially the PBE0 functional, for geometry optimizations. Adequate justification, including references to fundamental reasons or previous studies (including benchmark studies), supporting this choice of DFT functional, should be provided and explained in the text.

Response

We have added the following text and references supporting the choice of PBE0 functional and multireference ab initio methods for transition metal complexes to Section 4.2.

“The reason for using the PBE0 functional is its good performance in predicting molecular geometries of transition metal complexes [42-46]. The choice of the CASSCF and XMCQDPT2 approximations is due to the expected multireference nature of the wave functions for a number of considered electronic states.”

Comments 3 and 4

  1. It would be beneficial if the authors performed vibrational frequency calculations to confirm that the optimized geometries correspond to the true minima.
    4. The Cartesian coordinates of the optimized structure must be included in the Supporting Information file.

Response

Thank you for suggestion. Of course, we checked the nature of the stationary points by calculating the vibrational frequencies. The absence of imaginary vibrational frequencies confirmed the correspondence of the optimized structure to the minimum on the PES. The following information has been added to Section 2.3.

The atomic coordinates of the optimized structure and the vibration frequencies are given in the Supporting Information file (Tables S1, S2).

Reviewer 2 Report

It is a well written manuscript. All of the methods are plausible and applied with proper effort. I have only one some problem.

1. Can you guess how strong the Co-O bond (wieberg index : 0.28--0.3) compared to some other bond in similar type of salt (like Mg salt)

It is Ok.

Author Response

Dear Reviewer,

The authors of this manuscript are grateful to you for the comment. Unfortunately, we have no information about bond strength in complexes with such coordination of metal atom, including Mg, and could not trace some regularities of it. However, we studied several chelate complexes of some transition metals. We will collect such data and try to find the regularity for bond strength in series of metal complexes. Thank you for suggestion.

Please find below our responds including the table with discussed data.

Comment

Can you guess how strong the Co-O bond (wieberg index : 0.28--0.3) compared to some other bond in similar type of salt (like Mg salt)

Response

We have no information about correlation between length bond and Wiberg index for salts of Mg. But we have studied the variation of these parameters with the nature of some d-elements in several classes of coordination compounds. The table gives examples for Ni(II), Cu(II), and Zn(II) complexes.

Table. The NPA charges on the atoms q(M) in complexes of Ni(II), Cu(II) and Zn(II) and Wiberg index P of bonds

Co4O(piv)6

q(Co)= +1.20

Ni(acacen)

q(Ni)= +1.04

Cu(acacen)

q(Cu)= +1.34

Zn(acacen)

q(Zn)= +1.63

re

P

re

P

re

P

re

P

M–N

1.884

0.395

1.966

0.212

2.027

0.157

M–O

1.973-1.987

0.29

0.28

1.867

0.316

1.941

0.178

1.964

0.147

Ni(salen)

Cu(salen)

Zn(salen)

M–N

1.876

0.379

1.975

0.195

2.073

0.139

M–O

1.860

0.340

1.922

0.194

1.928

0.166

                           M(acacen)                                                                                          M(salen)

Some thoughts: Wiberg index P correlates with bond order, so P(OL-C) = 1.37 (Table 4, manuscript) shows that P of this bond is close to 1.5 and has a noticeable π-component.

In the Table the NPA charges on the atoms of the d-elements Ni(II), Cu(II) and Zn(II) are compared.  An increase in the positive charge on the M atom in the series Ni→Cu→Zn indicates an increase in the ionic component of M-O and M-N bonds. In this case, a decrease in the covalent component occurs, which is reflected in a decrease in the Wiberg index P.

The nature of the Co-O bond in Co4O(piv)6, as noted in the article, can be attributed to ionic (or coordination) bonds with a significant contribution of the covalent component, close to the nature of the Ni-O bond of Ni(acacen).

Reviewer 3 Report

The authors studied the gas phase structure of Oxopivalate Co(II). They found that the ferromagnetic and antiferromagnetic structures are close in energy, which indicate that the transformation between the different spin states can occur easily. The manuscript is well written and I only have a few observations.

I would like to know if the authors optimize the structure at the DFT level with other multiplicities besides 13. If so, how large were the changes in the geometry compared with M=13. At least M=1 should be optimized at the DFT level because this state was the one with the lower energy according to their CASSCF and XMCQDPT2 calculations.

Why the authors use the molecule in high spin for the QTAIM calculation, if the state M=1 is the one with the lower energy?

In the methods section, it is not indicated with which software and at which level of theory are the QTAIM calculation performed. Why the authors do not use the CASSCF wave function for QTAIM, it should be better and it is already calculated?

Author Response

Dear Reviewer,

The authors of this manuscript are grateful to you for your comments, which motivated us to improving the content and presentation of the manuscript. We tried to follow all your suggestions.

Please find below our responds.

  Comment 1

I would like to know if the authors optimize the structure at the DFT level with other multiplicities besides 13. If so, how large were the changes in the geometry compared with M=13. At least M=1 should be optimized at the DFT level because this state was the one with the lower energy according to their CASSCF and XMCQDPT2 calculations.

Response

Only high spin stated with multiplicity M=13 can be calculated by DFT with single-reference wave function. The CASSCF calculations showed that the states with M=1-11 possess a very complex composition of wave functions. We try to attract the intention of the readers to this fact by following text in Page 6 (Lines 203-206):

«These data, together with the results of the CASSCF calculation, show that it is impossible to single out any CSF with a dominant weight in the composition of the wave functions. The values of the coefficients with which CSF enter the wave functions of electronic states with M = 1 – 11 do not exceed 0.05.».

In these cases, the use of the DFT method based on a one-determinant wave function is excluded. Therefore, DFT calculations for states with M=1-11 were not performed. The calculations described in lines 206-220 showed that the geometry of the skeleton practically does not change during the transition from one electronic state to another.

 Comment 2

Why the authors use the molecule in high spin for the QTAIM calculation, if the state M=1 is the one with the lower energy?

Response

Unfortunately, the QTAIM program used by us can only works with a single-determinant wave function. Therefore, we were able to perform the QTAIM analysis only for the high-spin state. However, the similarity in geometrical parameters of complex and in orbital distribution electrons in different electronic states allows assuming the similar picture of QTAIM results.

Comment 3

In the methods section, it is not indicated with which software and at which level of theory are the QTAIM calculation performed. Why the authors do not use the CASSCF wave function for QTAIM, it should be better and it is already calculated?

 Response

Thank you for the hint on this our omission. In section “4.2. Details of quantum chemical calculation of Co4O(formyl)6 in different electronic states" we added the relevant text and reference.

“The topological analysis of electron density distribution function was carried out using AIMAll professional software [30].”
